# Remote blood pressure monitoring and behavioral intensification for stroke: A randomized controlled feasibility trial

**Beom Joon Kim**[1], **Jong-Moo Park**[2], **Tai Hwan Park**[3], **Joungsim Kim**[1], **JongShill Lee**[4], **Keon-Joo Lee**[1], **JiSung Lee**[5], **Jae Eun Chae**[6], **Lehana Thabane**[7,8], **Juneyoung Lee**[6], **Hee-Joon Bae**[1] *

1 Department of Neurology and Cerebrovascular Center, Seoul National University College of Medicine, Seoul National University Bundang Hospital, Seongnam, Republic of Korea, 2 Nowon Eulji Medical Center, Department of Neurology, Eulji University, Seoul, Republic of Korea, 3 Department of Neurology, Seoul Medical Center, Seoul, Republic of Korea, 4 Department of Biomedical Engineering, Hanyang University, Seoul, Republic of Korea, 5 Clinical Research Center, Asan Medical Center, Seoul, Republic of Korea, 6 Department of Biostatistics, College of Medicine, Korea University, Seoul, Republic of Korea, 7 Department of Health Research Methods, Evidence, and Impact, McMaster University, Hamilton, Ontario, Canada, 8 Biostatistics Unit, St. Joseph's Healthcare—Hamilton, Hamilton, Ontario, Canada

* braindoc@snu.ac.kr

**Data Availability Statement:** Anonymized dataset and data dictionary are attached as supporting information.

## Abstract

Measuring blood pressure (BP) at home and remote monitoring can improve the patient's adherence to BP control and vascular outcomes. This study evaluated the feasibility of a trial regarding the effects of an intensive mobile BP management strategy versus usual care in acute ischemic stroke patients. A feasibility-testing, randomized, open-labeled controlled trial was conducted. Remote BP measurement, data transmission, storage, and centralized monitoring system were organized through a Bluetooth-equipped sphygmomanometer paired to the participants' smartphones. Participants were randomized equally into intensive management (behavioral intensification to measure BP at home by texting, direct telephone call, or breakthrough visit) and control (usual care) groups. The primary feasibility outcomes were: 1) recruitment time for the pre-specified number of participants, 2) retention of participants, 3) frequency of breakthrough visit calls, 4) response to breakthrough visit call, and 5) proportions satisfying BP measurement criteria. Sixty participants were randomly assigned to the intensive management (n = 31) and control (n = 29) groups, of which 57 participants were included in the primary analysis with comparable baseline characteristics. Recruitment time from the first to the last participant was 350 days, and 95% of randomized participants completed the final visit (intensive, 94%; control, 98%). Eight breakthrough visit calls were made to 7 participants (23%), with complete and immediate responses within 3 ± 4 days. The median of half-day blocks fulfilling the BP measurement criteria per patient were 91% in the intensive group and 83% in the control group (difference, 12.2; 95% confidence interval, 2.2–22.2). No adverse events related to the trial procedures were reported. The intensive monitoring, including remote BP measurement, data transfer, and centralized monitoring system, engaged with behavioral intensification was feasible if the patients complied with the intervention. However, the device utilized would need further improvement prior to a large trial.

**Funding:** This trial was financially supported by a research fund from Daiichi-Sankyo Pharmaceuticals, but the sponsor had no role in design, data collection, analysis, interpretation and preparation of the manuscript.

**Competing interests:** This trial was financially supported by a research fund from Daiichi-Sankyo Pharmaceuticals, but the sponsor had no role in design, data collection, analysis, interpretation and preparation of the manuscript. This does not alter our adherence to PLOS ONE policies on sharing data and materials.

## Introduction

Achieving and maintaining the target blood pressure (BP) in an individual would be one of the top priorities in preventing further vascular events after an ischemic stroke [1]. Sophisticated BP control is now feasible with various antihypertensive medications in the market, but there are still unanswered questions related to BP management.

Most of the current scientific reports are based on the office-measured BP, which is allegedly higher than home-measured BP [2]. Home-measured BP would be a better indicator reflecting the levels and fluctuations of BP in daily life, but scientific hurdles still exist including measurement, transfer, monitoring, and interpretation of remotely assessed BP, as well as subsequent pharmacological modifications. Medication adherence is vital to maintain adequate BP control, but amost half of the hypertensive patients discontinue recommended medications within a year [3]. Moreover, clinical questions regarding BP variability would be answered better through remote collection of frequent home BP measurements than standard office BP measurements.

BP control is the most important intervention for preventing further vascular events in stroke patients, but majority of them have a certain degree of cognitive decline and functional disabilities against actively engaging in traditional instructions [4, 5]. Therefore, they are practical candidates of remote BP collection and monitoring strategy and continuous behavioral motivation with tele-health interventions. Recently, there are many smartphone-based mobile healthcare devices commercially available with the development of communication technology. Many companies offer Food and Drug Administration-cleared or EC Medical-certified wireless BP monitors for tele-medicine or user convenience [6].

Until now, the tele-health strategy encompassing remote BP measurements and collection and behavioral interventions has been tested separately in various social and medical conditions [7–11]. However, self-monitoring alone did not contribute to better BP control and additional behavioral interventions were necessary [12]. In this context, we designed a phase-II feasibility-testing, randomized, open, clinical trial for South Korean ischemic stroke patients. This trial aimed to verify the real-world feasibility of 1) home BP measurements using a Bluetooth-equipped sphygmomanometer paired to a patient-owned smartphone and transferring them to a central sever, and 2) of behavioral interventions for improving BP measurement and control in daily life, and breakthrough calls for any danger signs regarding BP levels and measurements.

## Methods

### Study design and participants

This study was a multicenter, randomized, open-label, prospective, phase-II feasibility trial to investigate the feasibility and safety of remote BP measurement and transfer system, based on a Bluetooth-equipped sphygmomanometer with centralized monitoring, behavioral intervention, and an antihypertension medication algorithm. Local Institutional Review Boards of recruiting centers approved the trial (SNUBH IRB#, B-1604/343-001) and all the study participants provided written informed consent voluntarily. This trial is registered at clinicaltrials. gov (NCT03024476). The authors confirm that all ongoing and related trials for this intervention are registered. Trial registration was completed after the first trial participant was enrolled due to clerical errors. The trial protocol is accessible at protocol.io (dx.doi.org/10.17504/ protocols.io.9nqh5dw). The first trial participant was enrolled on September 27, 2016 and the last participant completed the scheduled follow-up on December 7, 2017. The HEM-9200T, a Bluetooth-equipped sphygmomanometer used in the current trial, is a minor modification of

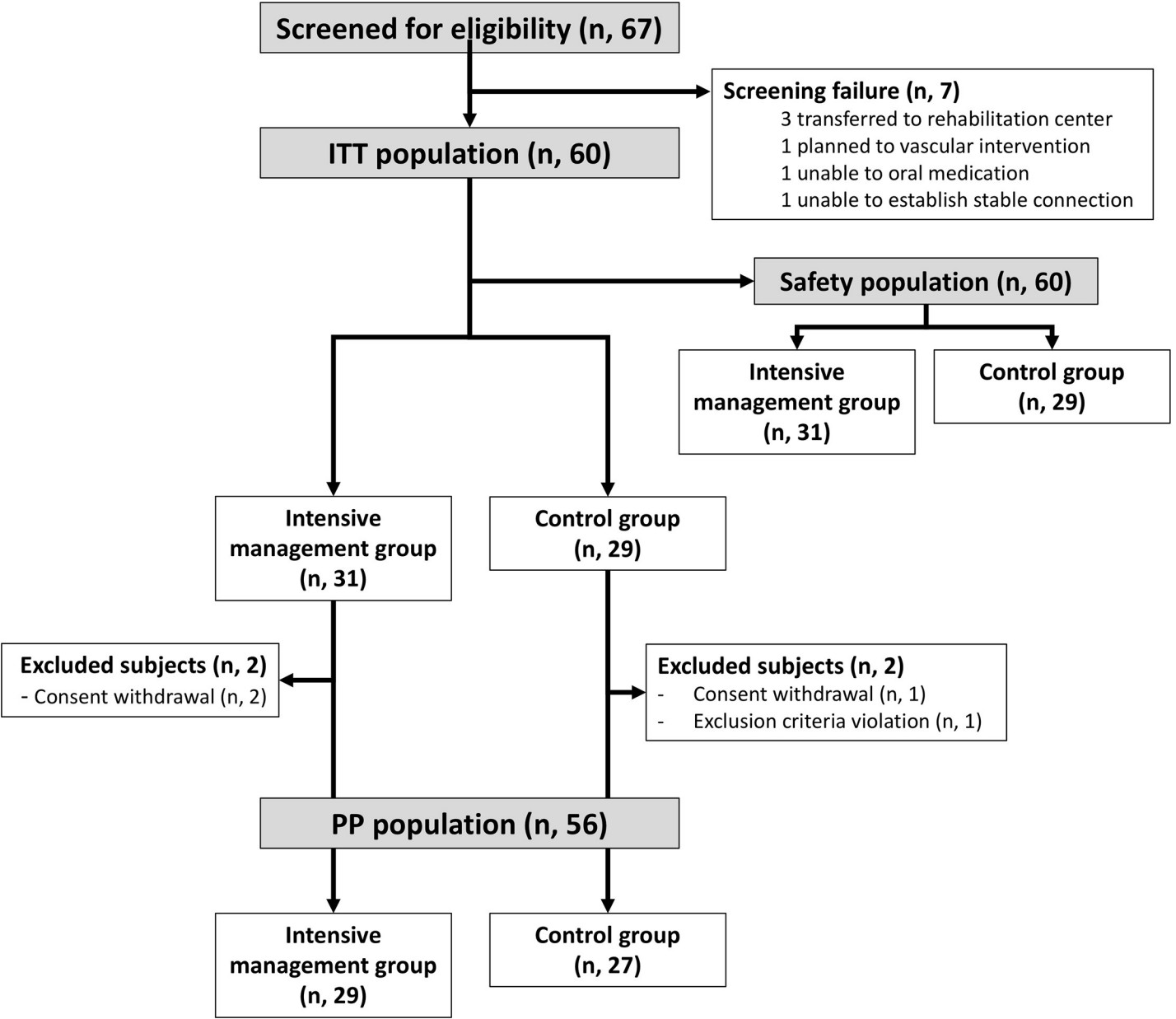

**Fig 1. Study profile and subject disposition.**

the currently marketed HEM-7311 cleared to market under 510 (k) K133379 with minor button changes and additional Bluetooth Low Energy communication function.

The inclusion criteria were 1) lesion-documented ischemic strokes, 2) age ≥19 years and admitted within 7 days after onset of symptoms, 3) mean systolic BP (SBP) ≥135 mm Hg for two consecutive days and >24 hours after onset of stroke, and 4) capability of using smartphones and a Bluetooth-equipped sphygmomanometer, and understanding trial instructions (Fig 1).

Major exclusion criteria were discharge to other facilities such as a nursing home or rehabilitation center, plans for endovascular interventions or vascular surgery within 3 months after stroke, or any known allergic reaction to olmesartan, amlodipine, or hydrochlorothiazide.

Details are provided in S1 Appendix. Among the ischemic stroke patients admitted in the participating centers in the trial period, 4.4% participants were randomized in the current trial (range, 3.1–6.4%).

## Randomization and trial procedures

Acute ischemic stroke patients with mean SBP $\geq$135 mm Hg during the two consecutive days, for >24 hours after onset of stroke, were screened for the trial. Trained research personnel instructed the patients for BP measurement, Bluetooth pairing between private smartphones and sphygmomanometer, and data transfer to the central server.

Screened patients were randomized at the time of discharge into the intensive management and control groups by a random number generator. Intensive management included behavioral intensification strategies such as 1) detailed instructions on regular measurement of BP ($\geq$5 days in a week, $\geq$2 times a day in the morning and evening, altogether $\geq$10 times during a week), 2) a short-message service (SMS) through their smartphones to encourage BP measurements as recommended when the participants failed to abide by instructions, and 3) a telephone contact by a trained research personnel and/or a request for a breakthrough visit when the number of BP measurements were $\leq$6/week or $\geq$50% of SBP measurements exceeded the pre-defined target range of 110–135 mmHg. Physicians were instructed to follow a pre-specified olmesartan-based prescription algorithm in managing trial participants of the intensive management group (S2 Appendix) [13, 14].

The control group participants were also provided a Bluetooth-equipped sphygmomanometer and instructed to pair it with their smartphones for data transfer. However, behavioral intensification, telephone contacts, breakthrough visit calls, and prescription algorithm were not applied. Although antihypertensive prescription in this group was at the discretion of the responsible physician, olmesartan was the first recommended medication.

Study participants were asked to visit the study clinics at 30 days (± 10 days) and 90 days (±14 days) after randomization. Trial researchers checked BP measurement, connection and transfer status of sphygmomanometer, and adherence to antihypertensive medication. Participants who were requested for a breakthrough call had a specified check-up at each unplanned visit (Fig 2).

## Measurement and transfer of home-measured BP

The trial organizer purchased a Bluetooth-equipped sphygmomanometer (HEM-9200T, Omron Healthcare, Co Ltd, Kyoto, Japan) and developed an Android application to establish a wireless connection between the smartphone and sphygmomanometer. The sphygmomanometer stored the BP measurements in the internal memory and automatically transferred the data to the in-house application installed on the paired smartphone. The application instantaneously transferred the BP data with date and time of measurements to the remote central server. Trial participants could see their BP data through the Android application and delete any erroneous values (Fig 3).

Trial personnel established and verified individual pairing between the trial participants' smartphones and the provided sphygmomanometer, and the data transfer from the smartphone to the central server during the screening period. One participant was not randomized due to insecure pairing, possibly due to technical problems of the smartphone (manufacturer, TCL mobile Ltd). After randomization, central trial personnel surveyed all the data transfer to the central server through a central monitoring system and issued a breakthrough call in case of sustained failure of the data transfer, and requested a visit to the outpatient clinics on the next business day. All the participants were asked to bring back the sphygmomanometer

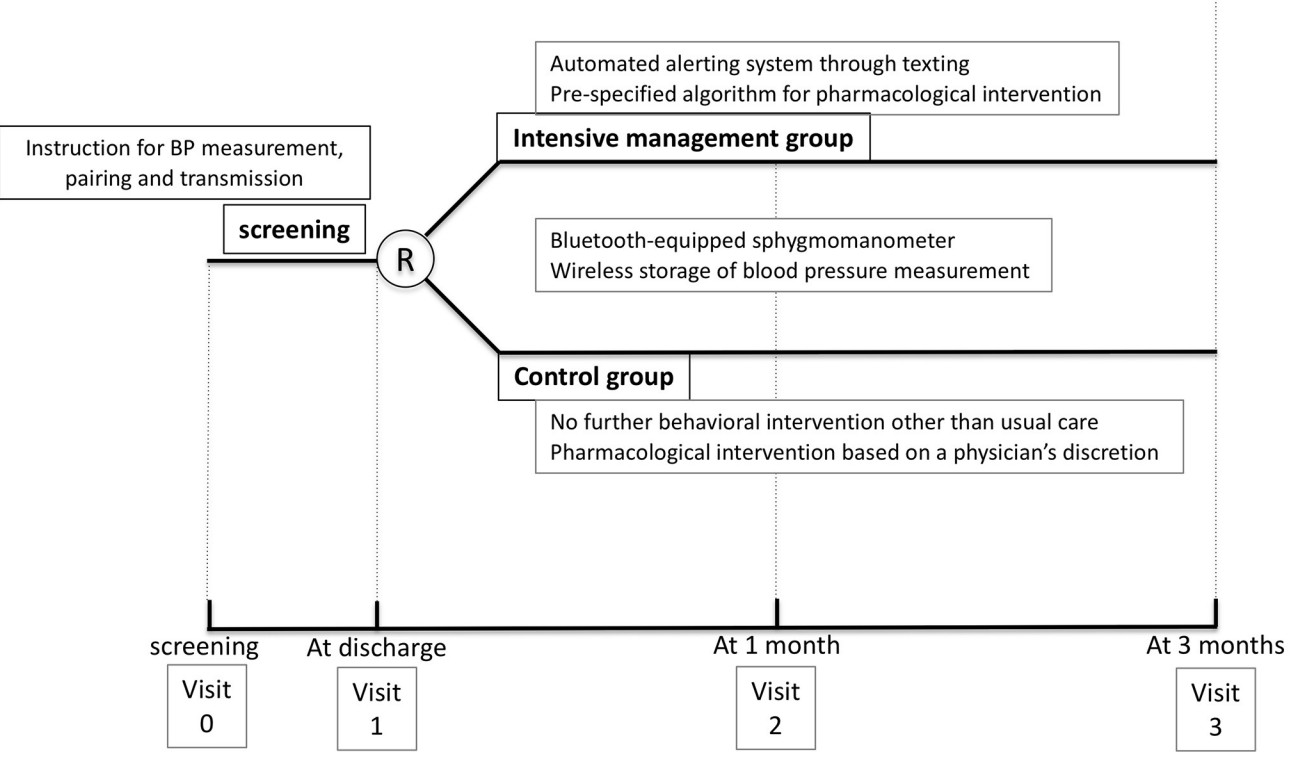

**Fig 2. Overall trial design.**

during the regular clinic visit. At the breakthrough visit, pairing between the smartphone and sphygmomanometer was checked; if this was not possible, then the BP data were registered manually. When data transmission failure occurred in the control group, the stored BP measurements during the failure period were downloaded directly from the device.

## Definitions of trial endpoints and criteria for success

As a phase-II trial, the primary feasibility endpoints of the current trial were as follows: 1) recruitment time of the pre-specified number of participants, 2) retention of included participants, 3) frequency of breakthrough visit calls, 4) number of patients responding to the breakthrough visit calls, and 5) proportion of patients fulfilling the criteria of BP measurement. Criteria for success are provided in Table 2 with the results of the primary endpoints. Secondary feasibility/efficacy endpoints were as follows: 1) average proportion of out-of-range (OOR) measurements, 2) weighted average proportion of OOR measurements (two-fold weights for consecutive OOR values), 3) vascular events including recurrent stroke, myocardial infarction, and any cause of death. Secondary safety endpoints included: 1) dizziness, fall, orthostatic hypotension, or any low BP-related events, 2) other adverse events potentially related to high or low BP, and 3) mortality (S3 Appendix).

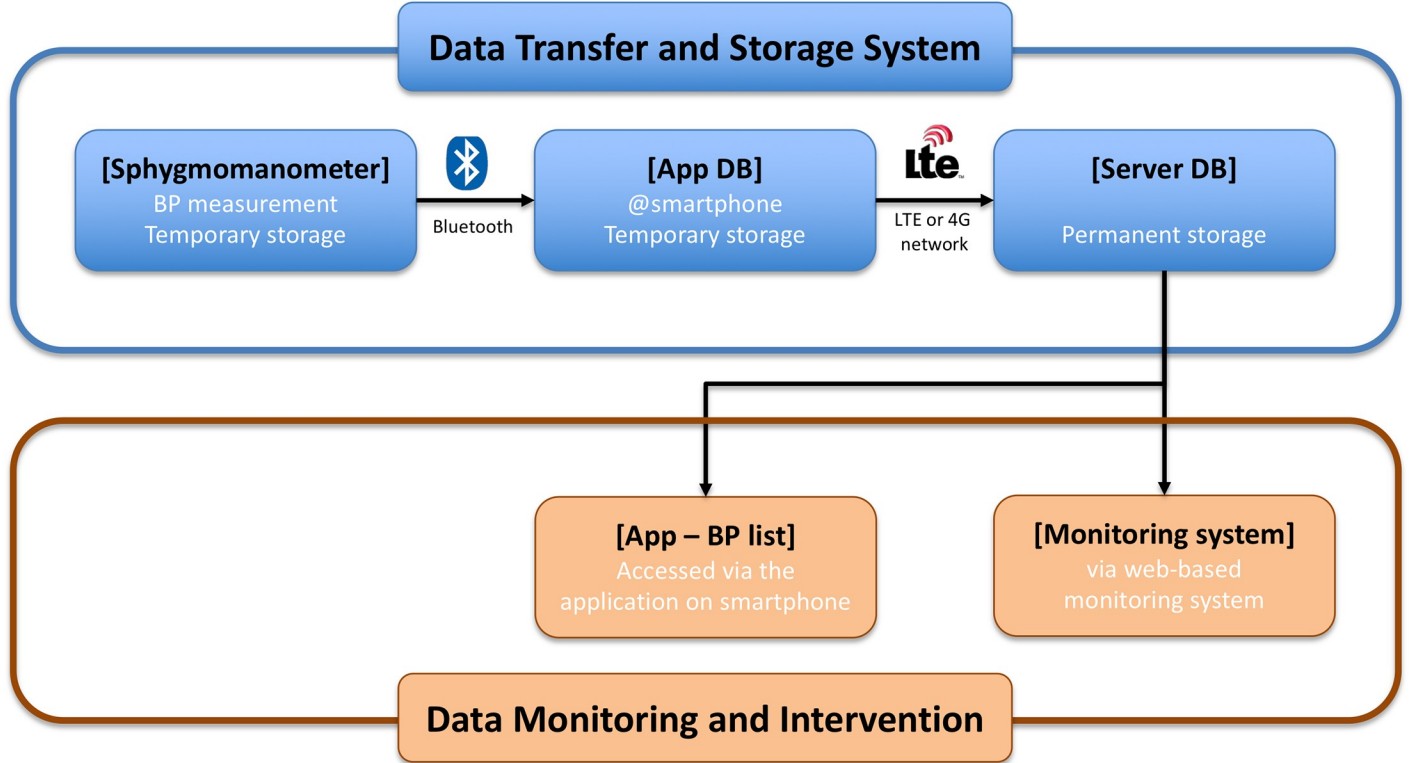

- BP measurement data are temporarily stored in the sphygmomanometer and the application in smartphone until transfer to the central server.
- Trial participants may check their own BP measurements through the application on smartphone.
- Trial registrar and research nurse may monitor individual BP profile and measurements through the monitoring system.

**Fig 3. Remote BP measurement, data transmission, storage, and centralized BP monitoring system.**

### Sample size

The sample size for the trial was based on feasibility considerations, because this trial focused primarily on determining feasibility of the main future trial. We aimed to recruit 60 patients who were randomized into two group (n = 30 per group). This number is within the recommended sample size for pilot or feasibility trials [15, 16].

### Statistical analyses

Feasibility or efficacy outcomes were evaluated using the intention-to-treat (ITT) as well as per-protocol (PP) analysis sets, and safety endpoints were examined using the safety analysis population. Demographics and baseline characteristics are presented by groups and are reported as mean (± standard deviation [SD]) or median (interquartile range) for continuous variables and count (percentage) for categorical variables. Feasibility or efficacy outcomes collected only from the intensive management group were summarized as numbers and percentages with 95% confidence intervals (CI). Mixed model for repeated measures analysis was performed to examine changes in mean SBP or diastolic BP (DBP) measurements across

**Table 1. Characteristics of the patients at baseline.**

| | Intensive management group (n, 31) | Control group (n, 29) |
|---|---|---|
| Male sex: n (%) | 19 (61%) | 20 (69%) |
| Age (yeas): Mean (SD) | 60 ± 12 | 56 ± 10 |
| Vascular risk factors: n (%) | | |
| Hypertension | 21 (68%) | 25 (86%) |
| Antihypertensive medication before stroke | 12 (39%) | 15 (52%) |
| Diabetes mellitus | 5 (16%) | 9 (31%) |
| Hyperlipidemia | 6 (19%) | 5 (17%) |
| Smoking | 12 (39%) | 11 (38%) |
| Atrial fibrillation | 2 (6%) | 4 (14%) |
| Stroke information | | |
| Stroke mechanism (TOAST): n(%) | | |
| Large artery atherosclerosis | 15 (48%) | 9 (31%) |
| Small vessel occlusion | 8 (26%) | 13 (45%) |
| Cardioembolism | 2 (6%) | 4 (14%) |
| Other determined etiology | 2 (6%) | 0 (0%) |
| Undetermined etiology | 4 (13%) | 3 (10%) |
| Baseline NIHSS score | 1 [0, 4] | 2 [1, 2] |
| Prestroke dependency (mRS score ≥1) | 2 (6%) | 2 (7%) |

follow-up weeks after randomization. In this analysis, a center effect was adjusted as a random factor. Due to a non-significant interaction between the study group and elapsed weeks from randomization, a test for equality as well as linear trend of mean SBP or DBP variables across weeks were performed for all participants (S2 Table).

All the statistical analyses were performed using the SAS 9.4 (SAS Institute, Cary, NC, USA) with a two-sided test, and statistical significance set at alpha = 0.05.

**Table 2. Results of primary endpoints and criteria for success of feasibility.**

| Primary endpoints | Intensive management group (n, 31) | Control group (n, 29) | Difference (95% CI) | Success criteria [1] | Decision for success |
|---|---|---|---|---|---|
| Recruitment time to prespecified number of subjects (days) | 328 | 340 | N/A | 10 months | Fail |
| Retention of included participants (n, %) | 29 (94%) | 28 (97%) | -3.00% (-14.21, 8.21) | 90% | Pass |
| Total number of frequency of calls for breakthrough visit (n) [2] | 8 | N/A | | 20 | Pass |
| Breakthrough visit response (n, %) | 8 (100%) | N/A | | 95% | Pass |
| Days between calls and visit (day) | 2 [0, 3] | | | 3 days | Pass |
| Compliance to BP measurements (n, %) | 31 (100%) | 26 (90%) | 10.34% (-1.36, 22.05) | 90% | Pass |
| Duration of transmission failure per subject (day) [3] | 0 [0, 0] | 0 [0, 1] | -4.53 (-9.72, 0.66) | 3 days | Pass |
| Percentage of half-day blocks satisfying BP measurement criteria per patient [4] | 91% [76, 97] | 83% [65, 90] | 12.22 (2.2, 22.24) | 80% | Pass |

Values presented as median [interquartile range], frequencies (percentages), or number.

1 Pass if the estimate exceeds the success criteria

2 Eight breakthrough visit calls were issued to 7 subjects (23%).

3 Mean ± SD, 0.7 ± 2.2 (intensive) versus 5.2 ± 13.5 (control)

4 P = 0.02 by Mann-Whiteney's U-test

## Results

Sixty ischemic stroke patients hospitalized in the three participating centers were randomly assigned to the intensive management group (n = 31; 52%) and the control group (n = 29; 48%). Three participants withdrew their consent during follow-up. One participant from the control group violated the exclusion criteria after randomization and was included in the ITT dataset, but removed from the PP analysis. Primary feasibility endpoints were collected for 57 participants (29 from the intensive management group and 28 from the control group).

The two groups were comparable for demographics, vascular risk factors, and stroke characteristics (Table 1). Hypertension was diagnosed in 46 participants (21 [68%] and 25 [86%] in the intensive management group and control group, respectively), and antihypertensive medication before the index stroke was prescribed in 27 participants (12 [39%] and 15 [52%] in the intensive management group and control group, respectively).

### Primary feasibility endpoints of the trial

Results of primary feasibility endpoints and their success criteria are presented in Table 2. Recruitment time from inclusion of the first trial participant to the final one was 350 days (S1 Fig). Among the randomized participants, 95% completed the final visit at 3 months after randomization (intensive group, n = 29 [94%]; control group, n = 28 [97%]. Eight breakthrough calls were requested to 7 participants (23%) in the intensive management group. Breakthrough calls were issued because of BP measurement transmission failure (n = 1.13%), elevated BP levels (n = 6.75%), and BP lower than the target range (n = 1.13%). Intervals from the breakthrough calls to visits were median 2 days (maximum 11 days). Results of the PP analyses are provided in the S2 Table.

The mean durations (±SD) of BP collection were 84.2 ± 20.0 days for the intensive management group and 88.7 ± 13.9 days for the control group. Transmission failure was observed in 5 participants (16%) of the intensive group and 8 (28%) of the control group, for mean (± SD) 0.7 ± 2.2 days in the intensive group and 5.2 ± 13.5 in the control group. The median (interquartile range) of blocks per patient that satisfied the criteria of BP measurements was 91% (76, 97) in the intensive group and 83% (65, 90) in the control group (difference, 12.2%; 95% CI, 2.2–22.2; P = 0.02 using Mann-Whitney's U-test; S1 Table). Mean BP levels were similar in both the groups over the trial period, irrespective of the ITT and PP population (Fig 4; S3 Table).

Of all the primary endpoints, only the recruitment time for the pre-specified number of patients failed to reach the success criteria for feasibility. The intensive management group passed the success criterion with respect to compliance of BP measurement, duration of transmission per participant, and proportions of half-day blocks per patient fulfilling the criteria of BP measurements, but the control group did not (S4 Appendix).

### Secondary feasibility and safety endpoints of the trial

The proportion of outliers, defined as a proportion of measurements being OOR for SBP values, was comparable between the two groups (intensive group, 0.4 ± 0.2%; control group, 0.4 ± 0.2%). Weighted proportion of SBP outliers was also similar (intensive group, 0.7 ± 0.3%; control group, 0.7 ± 0.4%). The occurrence of outliers above or below the recommended SBP range was not different (S5 Appendix) and rates of vascular events were comparable (Table 3).

Adverse events during the clinical trial were detected in 4 (13%) and 5 participants (17%) of the intensive management and control groups, respectively. In the intensive management group, no event related to the trial procedures was documented. In the control group, possible associations were suspected in one case of headache and one case of edema, but both of them

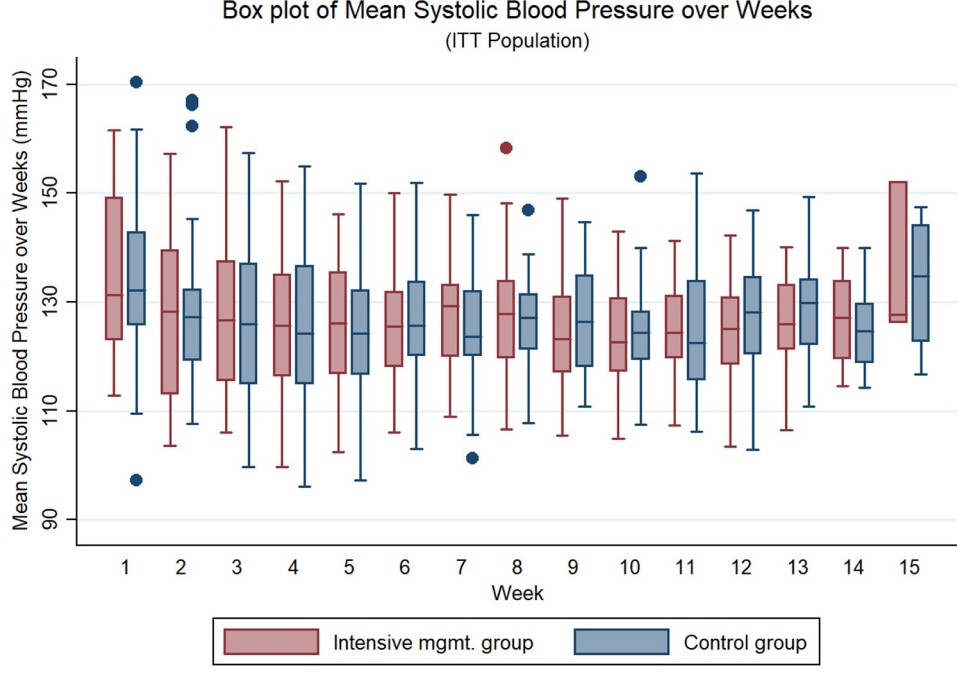

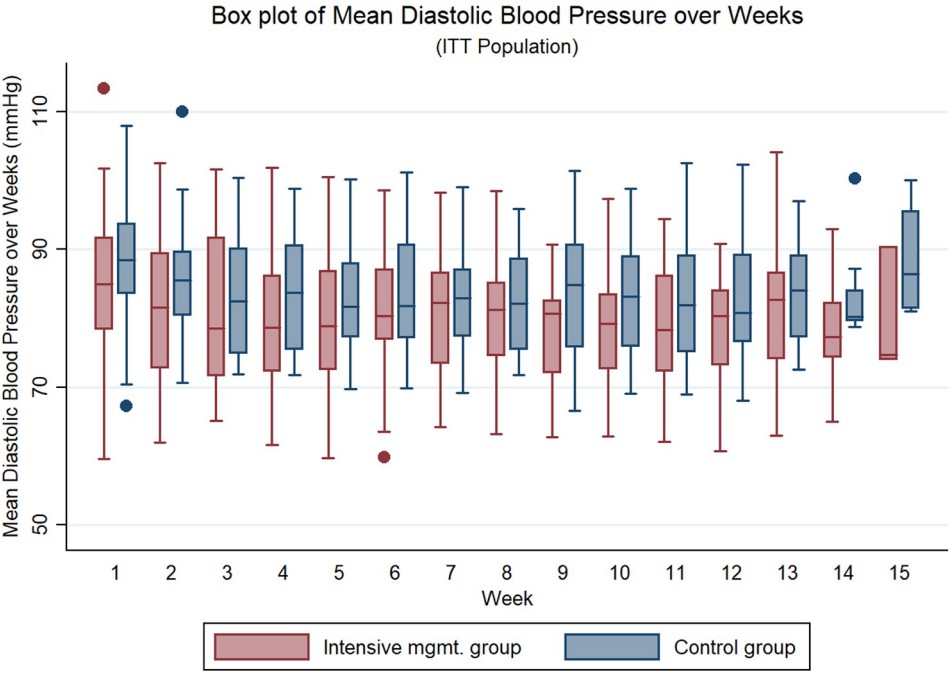

**Fig 4. BP measurements during trial period.**

**Table 3. Secondary outcomes of the trial.**

| | Intensive management group (n, 31) | Control group (n, 29) | Differences (95% CI) |
|---|---|---|---|
| Secondary efficacy endpoints | | | |
| Average proportion of OOR measurements (%) | 41 ± 17 | 43 ± 19 | -1.76 (-10.92, 7.41) |
| Weighted average proportion of OOR measurements | 66 ± 33 | 67 ± 35 | -0.58 (-18.06, 16.9) |
| Vascular events | | | |
| Recurrent stroke | 2 (6%) | 1 (3%) | 3.00% (-46.38, 52.38) |
| Myocardial infarction | 0 | 0 | |
| All kinds of death | 0 | 0 | |
| Secondary safety endpoint[†] | | | |
| Any adverse event | 4 (13%) | 5 (17%) | -4.34% (-50.98, 42.31) |
| Serious adverse event | 3 (10%) | 3 (10%) | -0.67% (-48.70, 47.36) |
| Mortality | 0 | 0 | |

Values presented as means ± standard deviations, frequencies (percentages), or medians [interquartile ranges].

were mild and the participants continued to participate in the trial. Serious adverse events were reported in 3 participants (10%) in the intensive group and 3 participants (10%) in the control group, but no relationship with the trial procedures was established (S4 Table). No participant was withdrawn from the trial due to occurrence of adverse events, and no mortality was reported.

Prescribed antihypertensive medications were well tolerated and compliance with the recommended algorithm was acceptable (S5 Table). Data on individual BP measurements are provided in S2 Fig.

## Reliable connection between the sphygmomanometer and the central system

Unexpected and unexplained technical failures occasionally occurred during the trial period. One participant was removed from the trial before randomization due to the unrecoverable failure in establishing Bluetooth pairing. After randomization, following incidences were reported and BP measurements had to be manually registered: unpaired device after Android operating system upgrade (n = 1), permanent failure in pairing of the sphygmomanometer and smartphones (n = 6), and intermittent unpairing but spontaneous recovery (n = 12). The sphygmomanometer was not portable and participants who travelled frequently had to record their BP manually. The trial participants were recommended to be the sole users of the Bluetooth-equipped sphygmomanometer, but there was no way to guarantee that.

## Discussion

For the current phase-II feasibility trial, we developed a wireless BP measurement, transmission, storage, and monitoring system using a Bluetooth-equipped sphygmomanometer paired to the participants' smartphones. Ischemic stroke patients were randomized into the intensive management and control groups to test whether this system with the behavioral intensification strategy for BP measurement and control was feasible in the real-world clinical practice. Therefore, we have documented that the inclusion and retention of trial participants proceeded as expected and the response rate of breakthrough visit calls was complete. The Bluetooth-based wireless BP collection system faced mechanical failure frequently and required centralized monitoring and technical interventions. Behavioral intensification, including SMS and telephone calls, increased the frequency of BP measurements. However, we did not

document any improvement in BP control in the intensive management group as compared to the control group.

We have proved the feasibility and safety of the wireless BP measurements, transmission, storage, and monitoring system based on a Bluetooth-equipped sphygmomanometer with behavioral intensification in ischemic stroke patients. Among the ischemic stroke admissions in the trial period, 4.4% patients were randomized in the current trial. Elevated BP levels are associated with increased risk of recurrent ischemic events in stroke patients [17]. Stricter control of hypertension can be started by increasing the frequency of BP measurements [18], and recently published guidelines recommend home BP monitoring and modification of antihypertensive medications accordingly [2]. Ischemic stroke patients with limited access to home BP monitoring, due to physical and cognitive disability, need additional attention in the provision of technical support [19, 20] for the application of our trial results. We used a pre-specified prescription algorithm of BP-lowering medication based on olmesartan, which was well tolerated. Additionally, BP control could be a feasible and practical area of tele-medicine [21]. With an emergent back-up system, such as breakthrough visit calls, Bluetooth-based data transfer and centralized monitoring system could work even for other health systems and disease entities.

Securing reliable connection for data transmission and storage between the trial participants and central server is a key component of the tele-health system [22]. The trial investigators tried to build a readily applicable technical system to collect, transfer, and store remotely measured BP data, and thereby adapted the Bluetooth technology that is prevalent and mature in the market [23, 24]. However, we experienced unexpected technical obstacles during the clinical trial, mostly due to improper functioning of the smartphone in conveying BP data to the central server. The smartphone application experienced one major upgrade and 72 minor modifications during the trial period. Future trials need to consider the instability of such a system, and try and establish a direct connection between the sphygmomanometer and data storage system via technology [25].

We issued breakthrough visit calls to trial participants in cases of data transmission failure and sustained BP outliers. Those calls received immediate and complete responses, suggesting that traditional medical interface would be a helpful back-up tool even in the era of tele-medicine. In this trial, there was no evidence regarding the effect of pre-specified BP-lowering medication algorithm, which could be due to the shorter trial duration. No adverse events related to this algorithm were reported.

The objective of the current trial was to prove the feasibility of remote BP collection/monitoring system and behavioral intensification strategy; hence, it was not adequately powered to document differences in BP control or clinical outcomes. The duration of our trial was limited to 90 days. We could not show any difference in BP control between the two groups. Since we utilized smartphones and Bluetooth-equipped sphygmomanometers, trial participants would have been younger, intellectual, and with milder stroke severity. Although we had randomized the participants, the control group may have been motivated to measure their BP more frequently. Providing a sphygmomanometer itself may have worked as a behavioral stimulus. We had to exclude severe stroke patients who were likely to need long-term care facilities. We cannot entirely exclude a chance of sharing the sphygmomanometer among family members although we explicitly discouraged that. The non-portability of BP monitors could discourage potential participants and manual recording of BP at home could be prone to errors. For future clinical trials, adoption of a user identification technology should be considered [26]. The trial comprised of ethnic and cultural Korean population, and the behavioral intervention and smartphone technology may not be applicable to other countries [27].

Several practical points are worthy of discussion for future trials or researches. Future implementation of a similar remote system should comply with the local regulations regarding security of data storage and transmission. The data server in this trial was physically separate from the hospital information system. Smartphones are widely used, but the elderly usually face difficulties in using and manipulating smartphones. Interfaces between smartphones and wireless healthcare devices, including BP monitors, could be difficult for them. To overcome the issue of technological illiteracy, a consumer-friendly user interface design such as iHealth Labs could be helpful.

## Conclusion

This phase-II randomized clinical trial proved the feasibility and safety of the intensive monitoring system, comprising of wireless home BP measurement, transfer, storage, and monitoring system and behavioral intervention, to improve BP measurement and control in acute ischemic stroke patients. However, technical considerations are important to establish secure connection for BP data transmission dependent on a smartphone. We also documented that the tele-medicine system still requires human interface to overcome technical failures and sustain behavioral changes for motivation. The tele-health system, as applied in the current trial, has not been perfected to the point that it did not require intense monitoring and adjustment by humans. Therefore, although the feasibility of the tele-health system in BP control of stroke patients has been documented and even if patients complied with the intervention, the system itself would certainly need to be perfected before any larger scale trials are considered.

## Supporting information

**S1 Checklist. CONSORT 2010 checklist of information to include when reporting a pilot or feasibility trial**∗**.**
(PDF)

**S2 Checklist. CONSORT 2010 checklist of information to include when reporting a pilot or feasibility randomized trial in a journal or conference abstract.**
(PDF)

**S1 Appendix. Detailed selection criteria.**
(PDF)

**S2 Appendix. Suggested prescription algorithm of blood pressure-lowering medication.**
(PDF)

**S3 Appendix. Details of trial endpoints: Half-day blocks and weighted average proportion of blood pressure measurement outliers.**
(PDF)

**S4 Appendix. Details on failure to reach success criteria for feasibility.**
(PDF)

**S5 Appendix. Blood pressure measurement outliers.**
(PDF)

**S1 Fig. Recruitment time to pre-specified number of participants by recruiting center.**
(PDF)

**S2 Fig. Individual blood pressure plot.**
(PDF)

**S1 File. Anonymized dataset and data dictionary.**
(XLSX)

**S1 Table. Proportion of patients fulfilling the criteria of blood pressure measurement.**
(PDF)

**S2 Table. Primary and secondary outcomes of the trial from per-protocol population.**
(PDF)

**S3 Table. Average blood pressure measurements during the trial.**
(PDF)

**S4 Table. Adverse events.**
(PDF)

**S5 Table. Compliance of study drugs.**
(PDF)

**S1 Protocol.**
(PDF)

**S2 Protocol.**
(PDF)

## Author Contributions

**Conceptualization:** Beom Joon Kim, Hee-Joon Bae.

**Data curation:** Beom Joon Kim, Jong-Moo Park, Tai Hwan Park, Hee-Joon Bae.

**Formal analysis:** JiSung Lee, Jae Eun Chae, Juneyoung Lee.

**Investigation:** Beom Joon Kim, Tai Hwan Park, Joungsim Kim, JongShill Lee, Keon-Joo Lee.

**Methodology:** JongShill Lee, JiSung Lee, Lehana Thabane, Juneyoung Lee, Hee-Joon Bae.

**Project administration:** Joungsim Kim.

**Resources:** Jong-Moo Park, Tai Hwan Park, Joungsim Kim, JongShill Lee.

**Software:** Keon-Joo Lee.

**Supervision:** Juneyoung Lee, Hee-Joon Bae.

**Validation:** Keon-Joo Lee.

**Visualization:** Keon-Joo Lee.

**Writing – original draft:** Beom Joon Kim.

**Writing – review & editing:** Jong-Moo Park, Tai Hwan Park, Lehana Thabane, Juneyoung Lee, Hee-Joon Bae.

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
