## [Decision Letter · Decision Letter 0]

22 Oct 2019

PONE-D-19-21291

Remote Blood Pressure Monitoring And Behavioral Intensification for Stroke: A Randomized Controlled Feasibility Trial

PLOS ONE

Dear Dr Kim,

Thank you for submitting your manuscript to PLOS ONE. After careful consideration, we feel that it has considerable merit but does not fully meet PLOS ONE’s publication criteria as it currently stands. Therefore, we invite you to submit a revised version of the manuscript that addresses the points raised during the review process. In particular, Reviewer 3 highlights potential issues that question if the system is really feasible at this point to justify a larger trial?The results to date might seem to suggest a need for further work?

We would appreciate receiving your revised manuscript by Dec 06 2019 11:59PM. To enhance the reproducibility of your results, we recommend that if applicable you deposit your laboratory protocols in protocols.io, where a protocol can be assigned its own identifier (DOI) such that it can be cited independently in the future. For instructions see: http://journals.plos.org/plosone/s/submission-guidelines#loc-laboratory-protocols

We look forward to receiving your revised manuscript.

Kind regards,

Antony Bayer

Academic Editor

PLOS ONE

Journal Requirements:

2) Please include captions for all your Supporting Information files at the end of your manuscript, and update any in-text citations to match accordingly. Please see our Supporting Information guidelines for more information: http://journals.plos.org/plosone/s/supporting-information.

3) Thank you for stating the following in the Competing Interests section:

[The authors have declared that no competing interests exist.].

We note that you received funding from a commercial source: Daiichi-Sankyo Pharmaceuticals

4)  We note that you have indicated that data from this study are available upon request. PLOS only allows data to be available upon request if there are legal or ethical restrictions on sharing data publicly. For information on unacceptable data access restrictions, please see http://journals.plos.org/plosone/s/data-availability#loc-unacceptable-data-access-restrictions.

Reviewers' comments:

Reviewer's Responses to Questions

**Comments to the Author**

1. Is the manuscript technically sound, and do the data support the conclusions?

Reviewer #1: Yes

Reviewer #2: Yes

Reviewer #3: Partly

2. Has the statistical analysis been performed appropriately and rigorously? 

Reviewer #1: Yes

Reviewer #2: Yes

Reviewer #3: N/A

3. Have the authors made all data underlying the findings in their manuscript fully available?

Reviewer #1: Yes

Reviewer #2: Yes

Reviewer #3: No

4. Is the manuscript presented in an intelligible fashion and written in standard English?

Reviewer #1: Yes

Reviewer #2: Yes

Reviewer #3: No

5. Review Comments to the Author

Reviewer #1: The authors have reported an excellent method for the remote BP measurement in ischemic stroke patients and demonstrated that it is feasible and safe. The study and its design is appropriate and the results have been well analysed.

The following minor suggestions are provided to make the manuscript more appealing for the readers.

1. The authors should add a relevant recent ref. for smartphone based mobile healthcare devices in the introduction.

Vashist, S. K., & Luong, J. H. T. (2019). Commercially Available Smartphone-Based Personalized Mobile Healthcare Technologies. In Point-of-Care Technologies Enabling Next-Generation Healthcare Monitoring and Management (pp. 81-115). Springer, Cham.

2. The authors should menton about the regulatory status of the Bluetooth-equipped sphygmomanometer (HEM-9200T, Omron Healthcare, Co Ltd, Kyoto, Japan) whether it is FDA and/or CE certified. Do the authors know whether the device has been checked for its equivalence to a clinically-validated predicate device for BP measurement?

3. If there are any limitations of the approach or the mobile healthcare technology involved, these need to be clearly reported in discussion. The most critical is the interface of the commercial mobile healthcare devices to the regulatory healthcare information systems (HIS) being used in healthcare. The second is the security of the data.

4. Some companies such as iHealth have developed dedicated softwares such as iHealth Pro in which they can integrate data from several mobile healthcare devices used by patients directly into a single software that can be easily viewed by doctor and easy to integrate with HIS. The authors should comment on this and discuss it in the discussion section.

5. The most significant advantage of mobile healthcare is the significantly reduced healthcare monitoring and management costs and more compliance expected from the patients together with their comfort and reduced time spent in clinics for follow ups. It would be useful if authors can comment on this imp aspect.

Reviewer #2: Authors present a report of a feasibility trial testing an intervention comprising of remote BP monitoring and behavioral intensification involving 60 ischemic stroke survivors. This report is well written and presents the data in a coherent manner. I wonder why it took a bit of time to reach sample size in reasonable time from 3 study sites.

Figure 4, week 15 systolic BP bar plot for the intensive group does not have confidence intervals showing in the figure.

No other comments

Reviewer #3: This study tested the feasibility of a trial for effects of a mobile BP management strategy vs. “usual care” for patients with acute ischemic stroke. Sixty participants were assigned randomly to intensive management or control groups. Five primary feasibility outcomes were examined, with 57 participants included in the primary analysis.

The authors are to be commended for presenting an exhaustive review of the trial findings; the study was obviously very well planned, and conducted and documented with great care. While the writing could benefit from careful review by a native English speaker to correct a number of minor grammatical errors, overall the results are clearly presented, and they are presented in an impressive amount of detail.

I do have some comments and suggestions that I believe would improve this manuscript; these are primarily related to the way certain aspects of the study are reported and the conclusions that have been reached.

1. The authors are actually varying two things in this study: (1) the monitoring system with or without fairly intensive support and (2) prescribed vs discretionary hypertension treatment. While the “control” group is receiving “usual care” in terms of being treated at the discretion of their physician, I am assuming that being given home monitoring equipment is not “usual” for ischemic stroke patients, so “usual care” isn’t really correct. This should be made more clear. Had a difference in BP management been observed, I’m not sure if the authors would have been able to differentiate the effects of intense monitoring from effects of differences in BP treatment. The authors should explain/justify why the treatment strategies were different for the two groups. This was either their decision, or they had no choice. Either way, this should be explained/justified.

2. Recruiting took a very long time, and did not meet the authors’ "success criteria." Since a larger trial would require an even larger cohort, it would be helpful if the authors discussed why they felt this was a problem and what they think might be done about it. The authors state that only 4.4% of patients with ischemic stroke during the time of the study were randomized, but the screening figure only gives information on those who were actually screened. Was the problem the fact that the eligibility criteria ruled out the majority of stroke patients, perhaps due to lack of technical ability? What percentage of the other 95.6% were actually eligible? How many eligible patients were approached and declined? How many would have been eligible except for a lack of technical skill? This information will be valuable in planning future recruiting efforts.

3. The authors state in the Results that “Unexpected and unexplained technical failures infrequently occurred during the trial period” but in the Discussion say “The Bluetooth-based wireless BP collection system faced mechanical failure not infrequently and required centralized monitoring and technical interventions.” Is the statement in the Results section incorrect?

4. It is not clear to me what happened if there was transmission failure in the control group. (Intervention group received breakthrough calls, but not the control group). The paragraph on technical failures does not differentiate by group; how were technical problems dealt with in the control group? Does this account for the difference in duration of transmission failure per subject/day (Table 2)?

5. It seems like the issue of non-portability is another limitation that should be discussed, as it could discourage potential subjects from wanting to participate. Having to record things manually while away from home would be prone to error.

6. I feel that the authors have focused the discussion section largely on their conclusion that they have proved feasibility; however, I wonder whether feasibility has really been proven. Based on the evidence presented, the authors have overstated the conclusions (i.e., proven feasibility) of conducting such a study on a larger scale. As the authors themselves state, “securing reliable connection…would be a key component. However, we have experienced unexpected technical obstacles…the smartphone application experienced one major upgrade and 72 minor modifications. Future trials need to consider the instability of such system.” This does not sound like a system that is “ready for prime time.” A more appropriate conclusion would be that, as designed, the system has not been perfected to the point that it did not require intense monitoring and adjustment. Therefore, the system itself would certainly need to be perfected before any larger scale trials could be considered. The authors conclude that BP control in the intensive management group did not improve over that in the control group. However, the performance from a technical standpoint WAS improved by intensive management. While the conclusion of “no difference” between intense and no management may be true for BP control, isn’t the more important point that intense management was needed to make the system work accurately at all? In the interest of performing future trials, I think that it would have been much more helpful if the discussion were focused on what didn’t work, and why, and how to fix it, rather than emphasizing the positive features of the trial and concluding that feasibility had been proven. There is a lot of valuable and important information here, but I don't think the authors have presented it in the most accurate or helpful light.

Minor comments:

• While the individual BP plots are beautiful and impressive, they really aren’t needed here, and take the focus away from the more important problems the investigators faced.

• The authors describe response to breakthrough times as “immediate”; however, in 4/7 cases, the response was delayed by 3 or more days, which in a short trial, could result in a sizeable loss of data. Again, be more realistic here; it is more helpful to look at the individual data and discuss where problems existed and why, than to make a judgment based on median delay which is indeed short.

• The percentage of half-day blocks reported for the intensive management group in Table 2 has a 95% confidence interval of 60% to 112%. This strongly implies that the data are not normally distributed and should not be summarized with a mean and sd or compared to a mean “success criteria.” A more meaningful standard would be to set the success criteria at or above a certain percentile (75th? 90th?), and evaluate the data with non-parametrics.

• This is also the case for duration of transmission failures. Please use non-parametric standards.

Overall, this is a very nice piece of work. Obviously a lot of time was spent summarizing a great deal of data quite beautifully. I think the study is very interesting, but I do feel like the authors failed in the discussion to really address the more interesting and more important aspects of their trial in sufficient detail. There were definitely some major problems in terms of feasibility, and the discussion would be much more valuable to future researchers (and realistic) if the authors addressed these. Information on why something ISN'T feasible is just as important (if not more so) than a (questionable) conclusion that it is.

6. PLOS authors have the option to publish the peer review history of their article (what does this mean?). If published, this will include your full peer review and any attached files.

Reviewer #1: No

Reviewer #2: Yes: Dr. Fred Stephen Sarfo

Reviewer #3: No

---

## [Author Response · Author response to Decision Letter 0]

4 Dec 2019

Respond to reviewers is prepared as a PDF format.

---

## [Decision Letter · Decision Letter 1]

27 Dec 2019

PONE-D-19-21291R1

Remote Blood Pressure Monitoring And Behavioral Intensification for Stroke: A Randomized Controlled Feasibility Trial

PLOS ONE

Dear Dr Kim,

Thank you for submitting your revised manuscript to PLOS ONE and for your careful attention to the reviewer comments. After careful consideration, we feel that it has merit but still does not fully meet PLOS ONE’s publication criteria as it currently stands. Specifically, I think Reviewer 3 makes a valid point about overstating your conclusion in both the abstract and main paper and you may think about using their suggested wording instead. The phrasing and grammar of the text would also benefit from attention and you may wish to get a native English speaker to revise it. Therefore, we invite you to submit a revised version of the manuscript that addresses these points. 

We would appreciate receiving your revised manuscript by Feb 10 2020 11:59PM. To enhance the reproducibility of your results, we recommend that if applicable you deposit your laboratory protocols in protocols.io, where a protocol can be assigned its own identifier (DOI) such that it can be cited independently in the future. For instructions see: http://journals.plos.org/plosone/s/submission-guidelines#loc-laboratory-protocols

We look forward to receiving your revised manuscript.

Kind regards,

Antony Bayer

Academic Editor

PLOS ONE

Reviewers' comments:

Reviewer's Responses to Questions

**Comments to the Author**

1. If the authors have adequately addressed your comments raised in a previous round of review and you feel that this manuscript is now acceptable for publication, you may indicate that here to bypass the “Comments to the Author” section, enter your conflict of interest statement in the “Confidential to Editor” section, and submit your "Accept" recommendation.

Reviewer #1: All comments have been addressed

Reviewer #3: (No Response)

2. Is the manuscript technically sound, and do the data support the conclusions?

Reviewer #1: Yes

Reviewer #3: Partly

3. Has the statistical analysis been performed appropriately and rigorously? 

Reviewer #1: Yes

Reviewer #3: N/A

4. Have the authors made all data underlying the findings in their manuscript fully available?

Reviewer #1: Yes

Reviewer #3: (No Response)

5. Is the manuscript presented in an intelligible fashion and written in standard English?

Reviewer #1: Yes

Reviewer #3: No

6. Review Comments to the Author

Reviewer #1: The authors have addressed all the review comments. The revised draft can be accepted for publication.

Reviewer #3: I thank the authors for their careful consideration of my comments and the manner in which they have been addressed. This is a small point, but I still feel that the conclusion as stated in the abstract and discussion (and conclusion) should be tempered to reflect that the authors have not proven feasibility with the tested system, as it is "not feasible" to give patients a device that requires so much maintenance. Perhaps a more cautious conclusion would be that intensive monitoring in itself was feasible, insofar as patients complied with the intervention, but that the device utilized would need to be improved prior to a larger trial.

Also, the paper still needs some work grammatically.

7. PLOS authors have the option to publish the peer review history of their article (what does this mean?). If published, this will include your full peer review and any attached files.

Reviewer #1: No

Reviewer #3: No

---

## [Author Response · Author response to Decision Letter 1]

29 Jan 2020

January 29th, 2019

Dear reviewers and editorial staffs of PLoS One;

We would like to thank the reviewers and editor for their positive evaluation and obliging suggestion on our manuscript. We provide responses to the reviewers’ comments below. Responses will follow the verbatim of reviewers’ comments (blue color). Additionally, we modified several sentences to make our point clearer. All the changes were specified in the marked version of our draft. 

Reviewer #3;

I thank the authors for their careful consideration of my comments and the manner in which they have been addressed. This is a small point, but I still feel that the conclusion as stated in the abstract and discussion (and conclusion) should be tempered to reflect that the authors have not proven feasibility with the tested system, as it is "not feasible" to give patients a device that requires so much maintenance. Perhaps a more cautious conclusion would be that intensive monitoring in itself was feasible, insofar as patients complied with the intervention, but that the device utilized would need to be improved prior to a larger trial.

Also, the paper still needs some work grammatically.

Editorial staffs;

Specifically, I think Reviewer 3 makes a valid point about overstating your conclusion in both the abstract and main paper and you may think about using their suggested wording instead. The phrasing and grammar of the text would also benefit from attention and you may wish to get a native English speaker to revise it. Therefore, we invite you to submit a revised version of the manuscript that addresses these points. 

We appreciate the reviewer #3's considerate suggestion on the conclusion and perspectives of our study results. After contemplation and along with the editor's advices, we concluded both abstract and main text with your wording.

Conclusion section of the abstract;

The intensive monitoring, including remote BP measurement, data transfer, and centralized monitoring system, engaged with behavioral intensification was feasible if the patients complied with the intervention. However, the device utilized would need further improvement prior to a large trial.

Conclusion section of the main text;

This phase-II randomized clinical trial proved the feasibility and safety of the intensive monitoring system, comprising of wireless home BP measurement, transfer, storage, and monitoring system and behavioral intervention, to improve BP measurement and control in acute ischemic stroke patients. However, technical considerations are important to establish secure connection for BP data transmission dependent on a smartphone. We also documented that the tele-medicine system still requires human interface to overcome technical failures and sustain behavioral changes for motivation. The tele-health system, as applied in the current trial, has not been perfected to the point that it did not require intense monitoring and adjustment by humans. Therefore, although the feasibility of the tele-health system in BP control of stroke patients has been documented and even if patients complied with the intervention, the system itself would certainly need to be perfected before any larger scale trials are considered.

---

## [Editor Report · Decision Letter 2]

10 Feb 2020

Remote Blood Pressure Monitoring And Behavioral Intensification for Stroke: A Randomized Controlled Feasibility Trial

PONE-D-19-21291R2

Dear Dr. Kim,

We are pleased to inform you that your manuscript has been judged scientifically suitable for publication and will be formally accepted for publication once it complies with all outstanding technical requirements.

With kind regards,

Antony Bayer

Academic Editor

PLOS ONE
---

## [Editor Report · Acceptance letter]

25 Feb 2020

PONE-D-19-21291R2 

Remote Blood Pressure Monitoring And Behavioral Intensification for Stroke: A Randomized Controlled Feasibility Trial 

Dear Dr. Kim:

I am pleased to inform you that your manuscript has been deemed suitable for publication in PLOS ONE. Congratulations! Your manuscript is now with our production department. 

With kind regards,

on behalf of

Professor Antony Bayer 

Academic Editor

PLOS ONE